# Analysis of Genetic Diversity of Fescue Populations from the Highlands of Bolivia Using EST-SSR Markers

**DOI:** 10.3390/genes13122311

**Published:** 2022-12-08

**Authors:** Karina Ustariz, Mulatu Geleta, Helena Persson Hovmalm, Rodomiro Ortiz

**Affiliations:** 1Centro de Investigación en Forrajes–La Violeta, Facultad de Ciencias Agrícolas y Pecuarias “Dr. Martín Cárdenas”, Universidad Mayor de San Simón, Cochabamba P.O. Box 4894, Bolivia; 2Department of Plant Breeding, Swedish University of Agricultural Sciences, P.O. Box 190, SE-23422 Lomma, Sweden

**Keywords:** Andes, Bolivia, fescue, genetic diversity, EST-SSR markers, microsatellites

## Abstract

In the highlands of Bolivia, native *Festuca* species are an important source of feed for animals due to their high tolerance to low temperatures and drought. Using simple sequence repeat (SSR) markers developed from expressed sequence tags (ESTs), the genetic diversity of 43 populations of *Festuca* species from Oruro, La Paz, Potosi and Cochabamba departments was evaluated for the purpose of providing information for effective conservation and breeding. In total, 64 alleles were detected across the 43 populations. SSR locus NFA 142 (with 12 alleles) had the highest number of detected alleles, while locus FES 13 (with eight alleles) had the highest polymorphism information content (PIC) at 0.55. Based on Nei’s genetic distance between populations, the unweighted pair group method with arithmetic mean (UPGMA) cluster analysis revealed two major clusters, each consisting of populations from the four departments. However, the analysis of molecular variance (AMOVA) revealed that only 5% of the total variation separated these two groups, indicating low genetic differentiation between the populations. It was also found that there was a low but significant differentiation (0.08%) between the population groups of the four departments (*p* = 0.01). The newly developed EST-SSR markers are highly valuable for evaluating the genetic diversity of Bolivian fescues and other related species.

## 1. Introduction

The genus *Festuca* L. belongs to the Poaceae family and is, along with the genus *Poa* L, the largest genus of the tribe Poeae. *Festuca* species are geographically distributed all over the world with a chromosome number ranging from diploid (2 n = 2 x = 14) to dodecaploid (2 n = 12 x = 84) [1]. According to the literature, the number of species ranges from 350 to 500 [2,3,4] and increases every year as new species are described. Genetic improvement of *Festuca* by conventional plant breeding is slow since many species are predominantly, if not completely, allogamous wind-pollinated grasses [5]. Although the diversity of *Festuca* is centered in the Holarctic zone of Eurasia and North America, approximately 140 species of this genus are found in South America [6], where the majority is growing in the high Andean zone of Peru and Bolivia. Of the 41 species of *Festuca* described in the Catalog of Vascular Plants of Bolivia [7], 17 are considered endemic to this region, and five have been reported to be in danger of extinction [8,9,10,11,12,13].

The most important feed sources for both camelids and ruminant species in the highlands are native grasslands, which are composed of different combinations of plant species (Chilliwar grassland, Pajonal, Tolar-*Parastrephia*-*Baccharis* grassland, Páramo rangelands and Bofedals) [14]. Natural prairies of native *Festuca* species are an important source of feed for animals inhabiting the highlands of Bolivia due to their high tolerance to low temperatures and drought. However, overgrazing, climate change, and the advance of the agricultural frontier have resulted in grassland reduction [15].

*Festuca dolichophylla* J. Presl (known as “chilliwa”) and *Festuca orthophylla* Pilg. (known as “iru ichu”) are the two most important forage species of this genus in the highlands of Bolivia [16,17]. The former constitutes a very important feed source for llamas, cattle and sheep [18,19,20], whereas the latter, despite being a poor forage due to its roughness and low nutritive value, is preferred by camelids, especially llamas [21,22]. Other *Festuca* species, such as *Festuca rigescens* J. Presl Kunth, *Festuca boliviana* E.B. Alexeev, and *Festuca humilior* Nees and Meyen, have also been reported as forage sources in the region [19].

Extensive research related to genetic diversity using different molecular markers such as simple sequence repeats (SSRs), amplified fragment length polymorphism (AFLP) and inter simple sequence repeats (ISSRs) has been conducted on *Festuca* species worldwide, especially in the two most economically important species, the hexaploid tall fescue (*Festuca arundinacea*) and the diploid meadow fescue (*Festuca pratensis*) [23,24,25,26]. However, there is still scarce information about the genetic diversity of fescues that originated in South America, including Bolivia, as a priority has been given to research on geographical distribution, taxonomy [19,27,28,29,30,31], nutritional content [20,32,33,34] and digestibility [35]. Even though the fescues are a very important source of feed for animals inhabiting this region, there is not much attention on issues regarding the conservation and improvement of native species in the Andean highlands. To the best of our knowledge, no molecular marker-based research on the genetic diversity of native fescues of Bolivia has been conducted. Information on the genetic diversity within and among populations is very important for the development of effective management strategies for plant species, particularly for endangered ones allowing effective conservation and genetic improvement in breeding programs.

Assessment of genetic diversity using molecular markers provides highly valuable basic information that can be utilized in breeding programs with the aim of developing suitable cultivars adapted to different environments and utilization systems [36]. Microsatellite or simple sequence repeats (SSRs) are widely used in genetic diversity assessment, “fingerprinting”, molecular mapping, and marker-aided breeding [37] due to their codominant inheritance, relative abundance, multi-allelic nature, extensive genome coverage, high reproducibility and use of simple methods for detection [38,39]. In addition, SSR loci have high rates of transferability across species within a genus [40,41,42,43,44]. The development of SSR from expressed sequence tag (EST) databases are a feasible option for obtaining high-quality DNA markers, which are transferable across related species [38,44,45,46]. In addition, EST-SSR is likely to be useful across a much broader taxonomic range as they derive from transcribed regions [47].

The main objective of this study was to evaluate the genetic diversity of 43 Bolivian populations of *Festuca* species using EST-SSR markers. The results are important for the effective management of native fescues in Bolivia, comprising both the development of conservation strategies and the future breeding of this forage grass.

## 2. Materials and Methods

### 2.1. Plant Material

Seeds of *Festuca* spp. were collected from 43 populations in the highlands of the departments of Oruro, La Paz, Potosi and Cochabamba from March to May 2015 (Figure 1; Appendix A). There were two separated ecozones represented by these populations, namely xerophytic and mesophytic Puna grasslands, situated at altitudes between 3217 and 4325 m above sea level (masl) [48,49]. Seeds were collected from 12 randomly chosen plants per population and germinated on filter paper in petri dishes at a mean temperature of 19 °C. Seedlings were transplanted and grown in the greenhouse of the Centro de Investigación en Forrajes (CIF)—La Violeta, Universidad Mayor de San Simón, Cochabamba, Bolivia. Young leaves were collected separately from all the 516 genotypes coming from the 12 plants per population, then they were placed in 2 mL Eppendorf tubes and lyophilized using a Labconco freeze dryer (FreeZone 6, model 77520, Kansas City, MO, USA) for 48 h. The freeze-dried samples were then sent to the Swedish University of Agricultural Sciences (SLU), Alnarp, Sweden, for DNA isolation.

### 2.2. DNA Isolation

To each freeze-dried sample, 2 glass beads (3 mm) were added, and then the samples were homogenized in a Mixer Mill MM400 (Retsch GmbH, Haan, North Rhine-Westphalia, Germany) for 1 min at 30 Hz. A total of 400 µL of lysis buffer (0.1 M Tris, 0.05 M EDTA, 1% SDS, pH = 9.0) was added, and the tube was then centrifuged at 13,000 rpm for 10 min. DNA was isolated from 200 µL of lysate by a QIAcube HT extraction robot using the QIAamp DNA mini kit (QIAGEN, Hilden, North Rhine-Westphalia, Germany) based on the standard protocol provided by the supplier with the addition of RNAse A having a final concentration of 0.1 mg/mL in the elution buffer. The quality and quantity of the isolated DNA were evaluated by agarose gel (1.5%) electrophoresis and spectrophotometry (Nano Drop ND-1000, Saveen Werner, Malmö, Sweden).

### 2.3. Designing and Screening Primer-Pairs

Based on the EST sequences of tall fescue available at the National Centre for Biotechnology Information (NCBI), sequences containing SSRs with two to six repeat motifs were screened using WebSat [50], a web software for microsatellite marker development (http://bioinfo.inf.ufg.br/websat/; accessed on 01 November 2022).

Once redundant, overlapping and very short sequences were excluded, 28 sequences containing SSRs were chosen as candidates for designing primer pairs using Primer3 primer designing program [51] (http://bioinfo.ut.ee/primer3-0.4.0/; accessed on 01 November 2022).

Twenty-eight newly designed primer pairs together with seven publicly available primer pairs: NFA036, NFA094, NFA126, NFA136, NFA142, NFA147 and NFA150 [33], were tested for amplification of their target genomic regions using DNA samples representing eight Bolivian fescue populations. Once the PCR conditions were optimized for each SSR locus, 12 primer pairs that consistently amplified their targets were selected for final analysis (Table 1). The forward primers of each of these primer pairs were 5′-labeled with either 6-FAM^TM^ or HEX^TM^ fluorescent dye.

### 2.4. PCR Amplification and Capillary Electrophoresis

PCR was performed in an S1000TM Thermal Cycler (Bio-Rad, Hercules, California, USA) in a reaction containing 2.5 µL PCR buffer (10 mM Tris-HCl, pH = 8.3 and 50 mM KCl), 0.3 mM of each dNTP, 0.3 mM of each forward and reverse primer, 1 U (0.04 U/µL) Dream Taq DNA Polymerase and 10 ng/µL of template DNA. Sterile Millipore water replacing DNA was used as a negative control. A 50 bp ladder (GeneRuler, Thermo Scientific, California, USA) was used to estimate the size of the amplified products. PCR products were analyzed using electrophoresis with 1.5% agarose gels containing GelRed™ and visualized using a UVP M-26XV benchtop UV transilluminator. Fragments were amplified using one of the following temperature profiles:(1)Initial denaturation step of 3 min at 95 °C, 38 cycles of 30 s denaturation at 94 °C, 30 s primer annealing at 60 °C, 45 s primer extension at 72 °C, and a final primer extension at 72 °C for 20 min (for NFA036).(2)Initial denaturation step of 3 min at 95 °C followed by nine touchdown cycles of 30 s denaturation at 94 °C, 30 s primer annealing at 58 °C reduced by 1 °C every cycle and 45 s extension at 72 °C. This was followed by 29 cycles of 30 s denaturation at 94 °C, 30 s annealing at 48 °C and 45 s primer extension at 72 °C with a 20 min final extension step at 72 °C (for FES09, FES14, NFA094, NFA142, NFA147 and NFA150).(3)Initial denaturation step of 3 min at 95 °C followed by nine touchdown cycles of 30 s denaturation at 94 °C, 30 s annealing at 59 °C reduced by 1 °C every cycle and 45 s primer extension at 72 °C. This was followed by 29 cycles of 30 s denaturation at 94 °C, 30 s annealing at 49 °C and 45 s primer extension at 72 °C with a 20 min final extension step at 72 °C (for FES04 and NFA126).(4)Initial denaturation step of 3 min at 95 °C followed by nine touchdown cycles of 30 s denaturation at 94 °C, 30 s annealing at 60 °C reduced by 1 °C every cycle and 45 s primer extension at 72 °C. This was followed by 29 cycles of 30 s denaturation at 94 °C, 30 s annealing at 50 °C and 45 s primer extension at 72 °C with a 20 min final extension step at 72 °C (for FES13, FES24 and NFA136).

Amplified products were kept at 4 °C until electrophoresis. The PCR products of the 12 EST-SSR loci were multiplexed into five panels based on their differences in fragment sizes or fluorescent labels of the forward primers, with each panel containing PCR products of two or three loci. For multiplexing the PCR products labeled with the same fluorescent dye, a minimum expected fragment size difference of 54 bp was considered to avoid overlapping. The multiplexed PCR products were diluted 25× using Millipore water, and the fragment analysis was performed on a 3500 Genetic analyzer (Thermo Fisher Scientific, Waltham, MA, USA).

### 2.5. Allele Scoring and Evaluation of Polymorphism

GeneMarker^®^ V2.7.0 software (SoftGenetics, LLC State College, PA, USA) was used for peak identification and allele size determination. Each peak was considered as an allele, and the genotype of each individual at each locus was determined and exported to Excel for statistical analysis. Observed number of alleles (Na), effective number of alleles (Ne), Nei’s gene diversity of each locus (H) [52], Shannon information index (I), total gene diversity (Ht), within-population gene diversity (Hs) and coefficient of gene differentiation (Gst) were calculated using POPGENE ver 1.32. [53].

To examine the genetic relationship among the 43 populations, Nei’s measure of genetic distance was calculated, and a dendrogram was constructed using the unweighted pair group method with arithmetic mean (UPGMA) using MEGA7 software [54,55]. To evaluate the differentiation among groups of populations, among populations and variation within populations, an analysis of molecular variance (AMOVA) was conducted using ARLEQUIN ver 3.5 software [56].

## 3. Results

### 3.1. EST SSR Polymorphism and Genetic Diversity

Of the 12 EST-SSRs, five were newly developed, and seven were previously published for tall fescue [38]. Their repeat motifs, expected sizes, observed size range and polymorphic information content (PIC) are provided in Table 1. Four of the loci (FES04, FES09, FES14 and NFA126,) had dinucleotide repeats, seven (FES13, FES24, NFA036, NFA094, NFA147, NFA142 and NFA150) trinucleotide repeats, and one (NFA136) tetranucleotide repeats. Two of the 12 loci (FES09 and NFA126) were monomorphic and, therefore, not considered for further analysis. The analysis of polymorphic information content (PIC) of the remaining 10 loci showed that FES13 and NFA142 were highly informative (PIC > 0.5), whereas FES24 and FES04 were moderately informative (PIC = 0.30 and 0.22, respectively). The remaining six loci were less informative, with PIC < 0.10 (Table 1).

In total, 64 alleles were recorded in the 10 polymorphic loci, with the observed number of alleles (Na) per locus ranging from three (NFA036 and NFA147) to twelve (NFA142), and an average of 6.4. The effective number of alleles (Ne) at each locus varied from 1.03 (NFA036) to 3.29 (NFA142) with an average of 2.0 (Table 2). The Nei’s gene diversity (H) varied from 0.03 (NFA036) to 0.70 (NFA142), with an average of 0.41. Other loci with high levels of gene diversity were FES13, FES24 and FES04, with a value of 0.66, 0.65 and 0.53, respectively. The Shannon diversity index (I) per locus ranged from 0.08 (NFA036) to 1.41 (NFA142), with an average of 0.80 across the loci (Table 2).

Allele frequencies for the 10 polymorphic loci ranged from 0.001 (NFA136, NFA142 and NFA150) to 0.987 (NFA036) (Table 3).

### 3.2. Within-Population Genetic Diversity

The percentage of polymorphic loci (PPL) in each population ranged from 40 to 90%, with a mean of 61.2% (Table 4). The highest PPL value (90%) was recorded for populations 5 and 8 (belonging to Oruro and Cochabamba, respectively), while the lowest PPL value (40%) was found in populations 15 and 23 (belonging to La Paz and Cochabamba, respectively) (Table 4, Appendix A). The average observed number of alleles (Na) per locus varied from 1.8 (population 23 from Cochabamba) to 3.6 (population 18 from La Paz). The corresponding effective number of alleles (Ne) per locus for these populations was 1.4 and 1.7, respectively. The mean values for the observed number of alleles (Na) and the effective number of alleles (Ne) across all loci and populations were 2.6 and 1.5, respectively. The Nei’s gene diversity (H) varied from 0.17 to 0.33 with an average of 0.24, whereas Shannon Index (I) varied from 0.29 to 0.63 with an average of 0.45 (Table 4). Among the 43 populations, populations 5, 6, 8, 18 and 34 were at the top in terms of diversity, with H above or equal to 0.31 and I above or equal to 0.60, representing Oruro, Cochabamba, La Paz and Potosi, respectively.

### 3.3. Genetic Variation among Populations and Groups of Populations

Analysis of molecular variance (AMOVA) using present/absent allele data of the 43 populations revealed no significant genetic variation among populations (*p* = 1; Table 5). Similarly, there was no significant differentiation among groups of populations when grouped according to their biogeographic provinces (FCT = 0.00022; *p* = 0.24). When populations were grouped according to departments, AMOVA allocated 0.08% of the total variation among groups indicating a very low level of genetic differentiation (FCT = 0.00082; *p* = 0.01; Table 5).

When AMOVA was calculated by grouping populations into two altitudinal ranges (populations sampled from altitudes below 3800 masl and populations sampled from altitudes above or equal to 3800 masl), no significant differentiation was found (*p* = 0.88). Therefore, the wide range of altitudes of the sample collecting sites (3217–4325 masl) did not influence the genetic variation among populations (Table 5).

### 3.4. Genetic Distance and Cluster Analysis

Nei’s genetic distance between populations ranged from zero (between populations 18 and 19 from La Paz) to 0.863 (between population 4 from Oruro and 36 from Cochabamba). The second lowest genetic distance (0.002) was recorded between population 8 from Cochabamba and 21 from La Paz. The second highest pairwise genetic distance (0.845) was observed between populations 4 and 38, followed by 0.831 (population 4 vs. 37) and 0.805 (population 26 vs. 36) (Appendix A).

Populations 36, 38 and 37 were the most distantly related populations, with mean genetic distances of 0.48, 0.47 and 0.46 from the other populations. The overall mean genetic distance between the populations was 0.31.

A UPGMA clustering with a bootstrap test was conducted based on Nei’s genetic distances at the population level. The 43 populations were grouped into two major clusters (I and II) supported by a moderate bootstrap value of 83% (Figure 2). Cluster I included populations 36, 37, 38, 39, 40, 41, 42 and 43 from Cochabamba; populations 29, 30, 31, 32, 33 and 34 from Potosi; populations 27 and 28 from La Paz; and population 35 from Oruro. Cluster II included populations 6, 7, 8, 9, 12, 22 and 23 from Cochabamba; populations 11, 13, 14, 15, 16, 17, 18, 19, 20, 21 and 26 from La Paz; and populations 1, 2, 3, 4, 5, 10, 24 and 25 from Oruro. Even though there was considerable intermixing, most of the populations from the same department were clustered together in the same major cluster. For example, all populations from Potosi were clustered in Cluster I (Figure 2). AMOVA revealed a significant variation (*p* < 0.05) between the two clusters, with 5% of the total variation differentiating them.

## 4. Discussion

Analysis of genetic diversity is not only important for crop improvement but also for efficient management and protection of the available genetic resources. EST-SSR markers have the advantage of facilitating population genetic analysis and are interesting because of their capacity to amplify conserved homologous sequences across different grass species [57], where some loci may be part of or linked to genes that regulate important phenotypic traits. Nonetheless, they are expected to exhibit lower levels of polymorphism than genomic SSRs. The results in this study showed that the EST-SSR loci NFA036, NFA094, NFA126, NFA136, NFA142, NFA147 and NFA150 developed for tall fescue [38] were transferable to Bolivian fescues, which are comprised of other *Festuca* species. These results are consistent with previous reports, which showed that EST-SSR markers have a high transferability rate among species of the same genus [38,40,41,42,43,58,59] or even higher taxonomic rank [60]. The locus NFA126 was monomorphic for the Bolivian fescues analyzed in this study, although it showed polymorphism in previous studies [38,58] of other grass species (including *Festuca* species). The average number of alleles per locus in the present study was 6.4, which is less than the value reported by Mian [58]. For example, nine alleles were recorded at locus NFA147 in their study, whereas only three alleles were identified in the present study, suggesting a relatively lower diversity in Bolivian fescues.

Among the five EST-SSR loci developed in the present study, FES09 was monomorphic, whereas FES04, FES13 and FES24 showed a relatively high level of polymorphism with PIC of 0.22, 0.30 and 0.55 and gene diversity (H) of 0.53, 0.65 and 0.66, respectively. Interestingly, the mean PIC of the newly developed polymorphic EST-SSRs (0.29) was higher than the mean PIC (0.12) of those polymorphic loci selected from Saha et al. [38] suggesting the advantage of using publicly available genomic resources for the development of novel highly informative molecular markers. Together with NFA142 (PIC = 0.52 and H = 0.70), the four newly developed polymorphic EST-SSRs are highly valuable for future population genetic analysis of fescues and other closely related species.

When the 43 populations were grouped according to their biogeographic provinces (xerophytic and mesophytic Puna), there was no significant differentiation among the groups indicating similar genetic diversity of Bolivian fescues across the two biogeographic provinces. The estimates of the within-population genetic diversity varied about twofold in both provinces, although the mean gene diversity (H) of xerophytic Puna (0.23) was slightly lower than that of mesophytic Puna (0.25). Both provinces also have a similar number of population-specific alleles (Table 3). The results suggest that it is equally likely to find fescue populations with a relatively high genetic diversity in both provinces. On the other hand, significant differentiation (*p* = 0.01) was obtained among accession groups that were grouped according to their departments (Cochabamba, La Paz, Oruro and Potosi), although it was very low. Among the four departments, the average gene diversity of populations from Cochabamba (H = 0.27) was slightly higher than that of the other departments (H = 0.23 to 0.24). A rare allele (at locus NFA-142) unique to Cochabamba and shared by only two populations (7 and 40) was also identified (Table 3). The results may suggest Cochabamba is a more suitable environment for in situ conservation of fescues in Bolivia. The within-population genetic diversity revealed in this study was lower than previous results obtained in similar EST-SSR or genomic SSR-based studies on natural populations of *Festuca* species [61] or other species [60,62]. A low genetic diversity among and within the Bolivian fescues obtained in this study is not in line with the statement made by Stancik and Peterson [63], who indicated that the South American Andes represent a significant center of *Festuca* diversity.

The absence of significant differentiation among populations, as revealed by AMOVA, strongly suggests the outcrossing nature of Bolivian Fescues that led to high gene flow between the populations. This is in line with a lack of a clear pattern of clustering of populations according to their geographic regions. For example, populations from Cochabamba were not clustered separately from those from La Paz, Oruro and Potosi, which can be attributable to a high gene flow between the regions. The fact that some populations from Cochabamba were closely clustered together with those from La Paz or Oruro or Potosi indicates a lack of significant correlation between genetic distance and geographical distance between the populations. Considering the absence of previous studies on native Bolivian fescues using molecular markers, the present results are important for the development of effective management strategies for native and endangered *Festuca* species, which will allow effective conservation and genetic improvement in breeding programs.

## 5. Conclusions

SSR loci FES04, FES13, FES24 and NFA142 can be considered suitable targets for future population genetic analysis in fescues or other Poaceae species owing to their high genetic polymorphism. In this study, no significant differentiation among populations was found, indicating a lack of clear population structure caused by factors such as high gene flow. The up twofold differences in genetic variation within populations obtained in this study have direct implications for setting conservation strategies, which include identifying genetic diversity hotspots for Bolivian fescues for in situ conservation. Although no clear genetic diversity hotspot was identified in the present study, the results suggest Cochabamba is a better-suited department for in situ conservation of Bolivian fescues. The fact that a significant differentiation (although very low) was found among population groups at a department level suggests the need for further studies that cover wider geographic areas to shed more light on the genetic diversity of Bolivian fescues and beyond.

## Figures and Tables

**Figure 1 genes-13-02311-f001:**
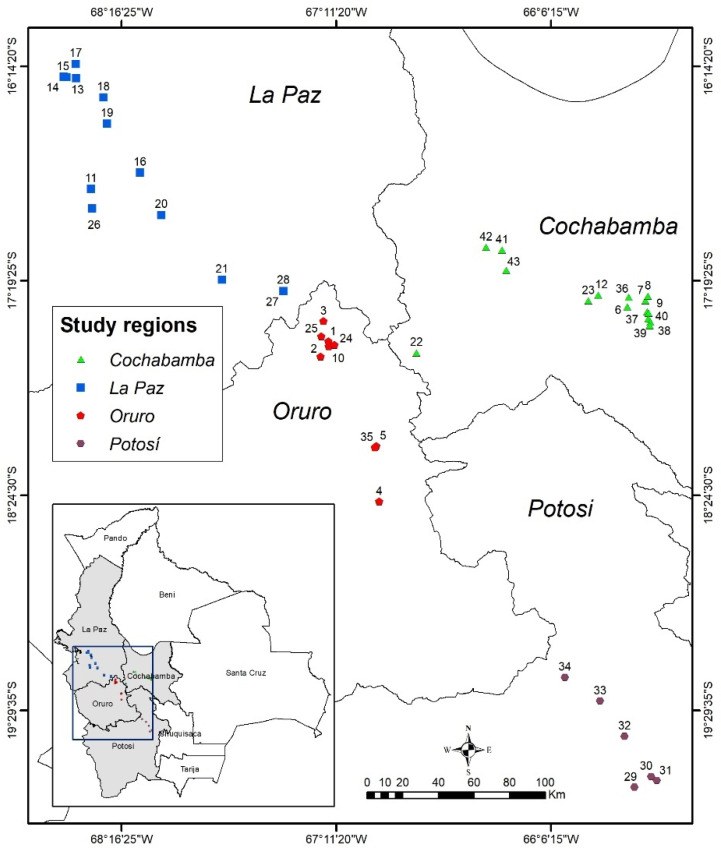
The map of Bolivia (bottom left) and its four departments showing locations where the 43 fescue populations analyzed in this study were collected.

**Figure 2 genes-13-02311-f002:**
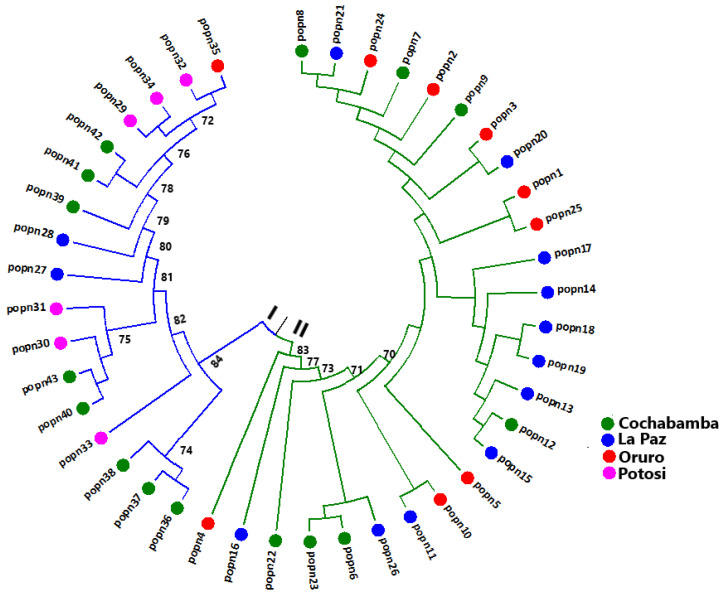
Unweighted pair-group method with arithmetic mean (UPGMA) tree showing the clustering pattern of the 43 populations based on Nei’s standard genetic distance. Numbers at the roots of the branches are bootstrap values. Note: Only bootstrap values ≥ 70% are shown.

**Table 1 genes-13-02311-t001:** Name and sequences of EST-SSR primer-pairs used in this study; repeat motifs, expected product size (bp), observed product size range (bp) and polymorphic information content (PIC) of their target loci.

Locus Name	Primer Sequence (5′–3′)	Repeat Motif	Expected Size	Observed Size Range	PIC
FES04 ^a^	F: GTTGAGTGGAGTTGAGGTCACA	(TA) 15	191	168–180	0.22
	R:AAAGGAACGAGGAAACACTACG				
FES09 ^a^	F: CCGGTGGGACCAGACTTA	(GC) 6	240	246	N/A
	R: CATGCACCCAGTTGTTGG				
FES13 ^a^	F: GCAATCTTTCCTTGCATCATCT	(TGC) 7	163	143–183	0.55
	R: AGCAGCCTCAACAACTCCAG				
FES14 ^a^	F: GCCCACACAACAAATGCAAC	(GT) 6	212	210–218	0.09
	R: CTCTCACTCTGCAGGTCGAT				
FES24 ^a^	F: CTTACACCAGGAAACGGAAGAT	(CAG) 7	251	234–255	0.30
	R: TGGAGAAGCAAAATGTGAAGTG				
NFA036 ^b^	F: AGAGGAAGAGCGAAAGAGCA	(GCC) 6	200	184–195	0.02
	R: CCCTGGTACTCGTGGATGTT				
NFA094 ^b^	F: AGCTGAACTATGAGGCATGTCA	(GCA) 6	247	234–261	0.04
	R: ATCCCTTTCCAGCATTTACCTC				
NFA126 ^b^	F: ACTACGTCTGCGAGTTCATTTG	(TA) 11	207	194	N/A
	R: GATCCACCGTTAGGAGAGTGTC				
NFA136 ^b^	F: TGTGCAAGCAGAGACCTACACT	(TGTT) 9	237	195–259	0.09
	R: CAGCTGTGCTCCATTATCTGAG				
NFA142 ^b^	F: CTTTGGACAAGGCAATGGAAT	(CAG) 7	229	210–255	0.52
	R: GTTGTTCTTCTGCGGGTAGTC				
NFA147 ^b^	F: TGCAGTCGGTTAAGATCAAGAA	(CTG) 7	215	201–209	0.03
	R: AGTTGCAGTGAAGGTGCTGAAC				
NFA150 ^b^	F: TGCAGTCGGTTAAGATCAAGAA	(CTG) 7	186	170–181	0.02
	R: GCAGAGCAATGGAGAGGTC				

^a^ = newly developed; ^b^ = previously published [39], N/A = not applicable.

**Table 2 genes-13-02311-t002:** Estimates of different genetic diversity parameters for the 10 polymorphic EST-SSR loci across the 43 Bolivian fescue populations.

Locus	Na	Ne	H	I	Ht	Hs	Gst
FES04	6	2.13	0.53	0.95	0.53	0.25	0.53
FES13	8	2.94	0.66	1.32	0.66	0.63	0.05
FES14	4	1.15	0.13	0.31	0.13	0.11	0.16
FES24	8	2.83	0.65	1.33	0.65	0.32	0.51
NFA036	3	1.03	0.03	0.08	0.03	0.02	0.10
NFA094	8	1.64	0.39	0.92	0.39	0.37	0.05
NFA136	8	2.06	0.51	0.89	0.52	0.09	0.83
NFA142	12	3.29	0.70	1.41	0.70	0.60	0.14
NFA147	3	1.04	0.04	0.12	0.04	0.04	0.11
NFA150	4	1.94	0.49	0.72	0.49	0.03	0.94
Mean	6.4	2.00	0.41	0.80	0.41	0.25	0.41
St. Dev.	2.9	0.82	0.26	0.49	0.07	0.05	

Na = observed number of alleles, Ne = effective number of alleles, H = gene diversity, I = Shannon information index, Ht = total gene diversity, Hs = within-population gene diversity, Gst = population differentiation, St Dev = standard deviation.

**Table 3 genes-13-02311-t003:** Alleles frequency distribution of the 10 polymorphic EST-SSR loci revealed in this study.

Allele	Locus									
	NFA036	NFA147	FES14	NFA150	FES04	FES13	FES24	NFA094	NFA136	NFA142
A	0.002 ^a^	0.010	0.010	0.011	0.304	0.029	0.327	0.007	0.019	0.003
B	0.987	0.979	0.032	0.380	0.004	0.046	0.036	0.040	0.015	0.002
C	0.012	0.011	0.933	0.609	0.612	0.431	0.014	0.051	0.011	0.029
D			0.024	0.001 ^b^	0.021	0.084	0.047	0.059	0.345	0.187
E					0.051	0.022	0.489	0.041	0.604	0.049
F					0.009	0.005	0.026	0.024	0.002	0.432
G						0.004	0.054	0.775	0.001 ^c^	0.007 ^h^
H						0.379	0.007	0.003	0.002 ^d^	0.281 ^e^
I										0.003
J										0.001 ^f^
K										0.003 ^g^
L										0.004

^a, b, c, d, e, f^ and ^g^ = specific alleles to populations 32, 8, 5, 20, 17, 20 and 4, respectively. Populations 8, 7 and 20 represent mesophytic Puna, whereas populations 4, 5 and 32 represent xerophtic Puna. ^h^ = specific alleles to populations 7 and 40 from Cochabamba representing mesophytic Puna.

**Table 4 genes-13-02311-t004:** Population diversity indices averaged over 10 polymorphic EST-SSR loci for each population.

Pop	NPL	PPL	Na	Ne	H	I		Pop	NPL	PPL	Na	Ne	H	I
1	8	80	2.5	1.4	0.22	0.4		23	4	40	1.8	1.4	0.18	0.31
2	5	50	2.4	1.4	0.21	0.38		24	5	50	2.3	1.4	0.22	0.39
3	5	50	2.7	1.6	0.23	0.44		25	5	50	2.4	1.6	0.24	0.44
4	7	70	3.2	1.7	0.26	0.53		26	5	50	2.4	1.6	0.24	0.44
5	9	90	3.1	1.8	0.33	0.63		27	6	60	2.6	1.5	0.22	0.41
6	8	80	3.1	1.8	0.33	0.61		28	6	60	2.5	1.5	0.22	0.41
7	7	70	2.8	1.6	0.28	0.52		29	6	60	3	1.4	0.24	0.47
8	9	90	3.1	1.7	0.33	0.6		30	5	50	2.2	1.3	0.18	0.32
9	6	60	2.4	1.6	0.27	0.48		31	6	60	2.4	1.4	0.21	0.38
10	5	50	2.6	1.5	0.22	0.41		32	5	50	2.2	1.4	0.21	0.38
11	7	70	2.8	1.5	0.24	0.47		33	6	60	2	1.4	0.21	0.36
12	8	80	2.4	1.7	0.31	0.53		34	8	80	3.3	1.7	0.33	0.63
13	5	50	2.9	1.5	0.25	0.47		35	5	50	2.3	1.5	0.2	0.38
14	6	60	2.6	1.5	0.24	0.46		36	7	70	2.1	1.6	0.27	0.45
15	4	40	2.4	1.4	0.2	0.37		37	7	70	2.6	1.7	0.3	0.54
16	5	50	2	1.3	0.17	0.29		38	7	70	2.6	1.5	0.25	0.46
17	5	50	2.1	1.4	0.17	0.31		39	6	60	2.3	1.7	0.29	0.49
18	8	80	3.6	1.7	0.31	0.63		40	6	60	2.4	1.4	0.22	0.4
19	8	80	3.3	1.7	0.28	0.58		41	5	50	2.8	1.6	0.27	0.51
20	6	60	2.6	1.5	0.24	0.44		42	6	60	2.3	1.6	0.26	0.46
21	5	50	2.6	1.5	0.23	0.42		43	6	60	2	1.4	0.22	0.36
22	5	50	2.5	1.5	0.22	0.42		Mean		61.2	2.6	1.5	0.24	0.45

Pop = population; NPL = number of polymorphic loci; PPL = percentage of polymorphic loci; Na = observed number of alleles; Ne = effective number of alleles; H = Nei’s gene diversity; I = Shannon information index.

**Table 5 genes-13-02311-t005:** AMOVA calculated for 43 populations (a) without grouping, (b) grouped into two biogeographic provinces (xerophytic and mesophytic Puna), (c) grouped into four departments (Oruro, La Paz, Potosi, Cochabamba) and (d) grouped into two altitudinal ranges (<3800 masl and ≥3800 masl).

Source of Variation	DF *	Sum of Squares	Variance Components	% Age of Variation	Fixation Indices	*p*-Value
*(a) Without grouping*						
Among populations	42	28.647	−0.43735 Va	−7.96	FST: −0.07962	1.00
Within populations	473	2805.000	5.93023 Vb	107.96		
Total	515	2833.647	5.49289			
*(b) By grouping the populations into two biogeographic provinces*						
Among groups	1	0.960	0.00121 Va	0.02	FCT: 0.00022	0.24
Among populations within groups	41	27.688	−0.43791 Vb	−7.97	FST: −0.07949	1.00
Within populations	473	2805.000	5.93023 Vc	107.95	FSC: −0.07973	1.00
Total	515	2833.647	5.49354			
*(c) By grouping the populations into four departments*						
Among groups	3	3.614	0.00452 Va	0.08	FCT: 0.00082	0.01
Among populations within groups	39	25.033	−0.44070 Vb	−8.02	FST: −0.07939	1.00
Within populations	473	2805.000	5.93023 Vc	107.94	FSC: −0.08028	1.00
Total	515	2833.647	5.49406			
*(d) By grouping the populations into two altitudinal ranges*						
Among groups	1	0.240	−0.00178 Va	−0.03	FCT: −0.00032	0.88
Among populations within groups	41	28.407	−0.43645 Vb	−7.95	FST: −0.07979	1.00
Within populations	473	2805.000	5.93023 Vc	107.98	FSC: −0.07944	1.00
Total	515	2833.647	5.49201			

## Data Availability

All the data is presented in the main text and Appendix A.

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
