# Peer review of "Analysis of Genetic Diversity of Fescue Populations from the Highlands of Bolivia Using EST-SSR Markers"

_genes, 2022, doi:10.3390/genes13122311_

Round 1

Reviewer 1 Report

The reviewed paper covers the topic of studying the genetic diversity of populations of Festuca species in Oruro, La Paz, Potosi, and Cochabamba. The study of the mentioned diversity was with the use of EST-SSR markers As for me, it is a niche species, although the authors emphasize the fact that in Bolivia it is very important as food for animals. The material and methods were correctly presented. Tables and graphs are clear. The methodology including AMOVA and cluster analysis is quite poor. It allowed us to test the research hypothesis, but there is no causal explanation.  I have mixed feelings about the innovation of this research. The results obtained are also not very revealing. I would suggest digging deeper into the hypothesis and statistical methodology to explain the cause of the results, perhaps with regression analysis. 

Note- statistically significant differentiation is the wrong formulation, significant differentiation is enough. 

Note 2- p-value level is usually presented to two decimal places. Consistency in the results presented should be demonstrated. 

Note 3- In conclusions, results should not be repeated. 

Reviewer 2 Report

Material and method section should be supported with references.

Reviewer 3 Report

See my comments in attached pdf. There were 12 genotypes in each population. indication 516 genotypes were studied. Author should also perform structure analysis for population structuring. 
